# Long-Term Effect of Combination of Creatine Monohydrate Plus β-Hydroxy β-Methylbutyrate (HMB) on Exercise-Induced Muscle Damage and Anabolic/Catabolic Hormones in Elite Male Endurance Athletes

**DOI:** 10.3390/biom10010140

**Published:** 2020-01-15

**Authors:** Julen Fernández-Landa, Diego Fernández-Lázaro, Julio Calleja-González, Alberto Caballero-García, Alfredo Córdova, Patxi León-Guereño, Juan Mielgo-Ayuso

**Affiliations:** 1Laboratory of Human Performance, Department of Physical Education and Sport, Faculty of Education, Sport Section, University of the Basque Country, 01007 Vitoria, Spain; julenfdl@hotmail.com (J.F.-L.); julio.calleja.gonzalez@gmail.com (J.C.-G.); 2Department of Cellular Biology, Histology and Pharmacology. Faculty of Health Sciences, University of Valladolid. Campus de Soria, 42003 Soria, Spain; diego.fernandez.lazaro@uva.es; 3Department of Anatomy and Radiology. Faculty of Health Sciences, University of Valladolid, Campus de Soria, 42003 Soria, Spain; albcab@ah.uva.es; 4Department of Biochemistry, Molecular Biology and Physiology, Faculty of Health Sciences, Campus de Soria, University of Valladolid, 42003 Soria, Spain; a.cordova@bio.uva.es; 5Faculty of Psychology and Education, University of Deusto, Campus of Donostia-San Sebastián, 20012 San Sebastián, Guipúzcoa, Spain; patxi.leon@deusto.es

**Keywords:** muscle recovery, exercise-induced muscle damage, lactate dehydrogenase, creatine kinase, testosterone, cortisol, sport nutrition, supplementation

## Abstract

Creatine monohydrate (CrM) and β-hydroxy β-methylbutyrate (HMB) are widely studied ergogenic aids. However, both supplements are usually studied in an isolated manner. The few studies that have investigated the effect of combining both supplements on exercise-induced muscle damage (EIMD) and hormone status have reported controversial results. Therefore, the main purpose of this study was to determine the effect and degree of potentiation of 10 weeks of CrM plus HMB supplementation on EIMD and anabolic/catabolic hormones. This study was a double-blind, placebo-controlled trial where participants (*n* = 28) were randomized into four different groups: placebo group (PLG; *n* = 7), CrM group (CrMG; 0.04 g/kg/day of CrM; *n* = 7), HMB group (HMBG; 3 g/day of HMB; *n* = 7), and CrM-HMB group (CrM-HMBG; 0.04 g/kg/day of CrM plus 3 g/day of HMB; *n* = 7). Before (baseline, T1) and after 10 weeks of supplementation (T2), blood samples were collected from all rowers. There were no significant differences in the EIMD markers (aspartate aminotransferase, lactate dehydrogenase, and creatine kinase) among groups. However, we observed significant differences in CrM-HMBG with respect to PLG, CrMG, and HMBG on testosterone (*p* = 0.006; η^2^p = 0.454) and the testosterone/cortisol ratio (T/C; *p* = 0.032; η^2^p = 0.349). Moreover, we found a synergistic effect of combined supplementation on testosterone (CrM-HMBG = −63.85% vs. CrMG + HMBG = −37.89%) and T/C (CrM-HMBG = 680% vs. CrMG + HMBG = 57.68%) and an antagonistic effect on cortisol (CrM-HMBG = 131.55% vs. CrMG + HMBG = 389.99%). In summary, the combination of CrM plus HMB showed an increase in testosterone and T/C compared with the other groups after 10 weeks of supplementation. Moreover, this combination presented a synergistic effect on testosterone and T/C and an antagonistic effect on cortisol compared with the sum of individual or isolated supplementation.

## 1. Introduction

Elite sports are immensely demanding at the physical and psychological level, and an adequate balance between training load and recovery is needed to achieve maximum performance [1]. Prolonged and excessive training and/or insufficient recovery can result in nonfunctional overreaching or an overtraining syndrome due to increased stress and fatigue and, consequently, impairment or stagnation of performance [1,2,3]. Therefore, recovery processes are determinants in sport performance [4]. In that way, there are different markers to determine stress and fatigue levels due to the imbalance between training load and recovery, among which biochemical markers of exercise-induced muscle damage (EIMD) and anabolic/catabolic hormones are often used [5].

In particular, an increase in EIMD markers produced by sarcomeric degeneration from Z-disk fragmentation [6,7], such as creatine kinase (CK) and lactate dehydrogenase (LDH), may provide information about elevated muscular and/or metabolic stress [1,8]. Further, the peak of CK, which presents the breakpoint between 300 IU/L and 500 IU/L, can increase until 24–96 h after exercise (up to four-fold rest values) [6,9]. Likewise, LDH can increase until 3–5 days (up to two-fold rest values) [6] depending on different types of exercise/sport [6]. Despite the increase in EIMD being dependent on the training level of athletes, the intensity and duration of the exercise and the density and type of sport are also important [6]. In particular, eccentric and impact activities produce more significant EIMD than concentric and nonimpact activities [6].

On the other hand, testosterone and cortisol hormones, such as anabolic and catabolic hormones, are highly useful for monitoring training adaptation [10]. However, chronic hormone concentrations, not acute, are most suggestive of long-term performance changes [11,12]. Testosterone is an androgenic and anabolic hormone secreted by the hypothalamus–pituitary–testicular axis, and an increase indicates an anabolic state [10,13,14]. Likewise, cortisol is a steroid hormone considered to be an indicative factor of accumulated stress intensity, which is secreted by the hypothalamus–pituitary–adrenal axis, and an increase of it indicates stress accumulation (catabolic state) [10,13,14]. Therefore, an increase in testosterone and a decrease in the resting plasma testosterone/cortisol ratio (T/C) has been proposed as an indicator of adaptation to training, indicating that an increase in T/C shows better adaptation [15,16].

In order to reduce EIMD and modify testosterone and T/C, as indicators of better recovery status [1,17,18], different ergogenic aids, such as creatine monohydrate (CrM) and β-hydroxy β-methylbutyrate (HMB), have been proposed [19,20]. In this sense, CrM has been shown to potentially reduce CK [21,22,23], LDH [21,23], and cortisol [24] levels and increase testosterone levels [24,25]. The pathway by which CrM may prevent or delay fatigue, reduce EIMD, and improve anabolic/catabolic hormones is that CrM supplementation increases muscle phosphocreatine (PCr) storage [26], and therefore, fatigue-produced muscle degeneration can be prevented or delayed by the CK/PCr system. Likewise, CrM can enhance skeletal muscle glycogen storage [27,28], although the main reason for this increase is unclear [27]. On the other hand, HMB has presented decreases in CK [29,30,31,32], LDH [31], and cortisol [32,33,34] levels and increases in testosterone [35]. The pathway by which HMB could influence the delay of fatigue could be through an enhancement of glycogen synthesis [36] by the inhibition of glycogen synthase kinase-3 (GSK3), which negatively regulates glycogen synthesis [37]. Moreover, HMB can increase protein synthesis by the stimulation of insulin-like growth factor-1 (IGF-1), which stimulates the phosphatidylinositol 3-kinase pathway (PI3K/Akt) pathway that activates mammalian target of rapamycin (mTOR) phosphorylation [38].

Although the effects of supplements are usually investigated in isolation, it is very common to combine different ergogenic aids in the sports field in order to achieve maximum performance [39,40,41]. In this sense, it could be considered that the effect of this combination might be better than individual intake. This hypothesis could be supported given that CrM and HMB (both supplements) have shown that they may act in common and in different physiological pathways related to EIMD and anabolic/catabolic hormones status [26,42]. However, it is not clear what the interaction effect (synergistic, multiplicative, antagonistic or nullifying) of multiple treatments given to athletes are [43]. Concretely, CrM plus HMB supplementation has not shown better results than isolated supplementation (CrM alone or HMB alone) on EIMD markers [44,45,46,47] or on anabolic/catabolic hormone status [45]. These results could have been influenced by blood sampling timing [46,47] when the main aim was to determine the long-term effect of CrM plus HMB supplementation and there being insufficient time to avoid prior training effects. Likewise, these controversial results could have been predisposed by short-term treatment (from six days to six weeks) [44,45,46] to achieve particularly an exercise-induced increase in hormone status [11,12].

Thus, the main purpose of this study was to determine the effect and degree of potentiation of long-term (10 weeks) mixing of 3 g/day of HMB plus 0.04 g/kg/day of CrM on EIMD markers (aspartate aminotransferase (AST), LDH, and CK) and anabolic/catabolic hormone status (testosterone, cortisol, and T/C) in elite male traditional rowers during the competitive period. We hypothesized that the mixture of HMB plus CrM could reduce EIMD and enhance endogenous recovery and exercise adaptations more than CRM or HMB separately.

## 2. Material and Methods

### 2.1. Participants

Twenty-eight elite male traditional rowers (30.43 ± 4.65 years and 59.92 mL/min/kg of VO_2 max_) belonging to one of the 12 teams that make up the rowing First Trainer League in Spain (ACT) participated in this study (22 rowers from the first team and 6 from the reserve team). Although some rowers had minor injuries and missed some training sessions, all of them completed the entire study rigorously without any drop-out. These rowers performed the same loads and number of training sessions with a daily duration of 1.5 h/day, 6 days/week for 10 weeks during the in-season period (competitive period). Thus, a personal diet was developed for each participant by the certified nutritionist of the rowing club. The diets for adequate athletic performance were suggested using energy and macronutrient guidelines [48] considering the training volume, training load, and individual characteristics of each rower.

All athletes completed a medical history questionnaire and electrocardiographic and cardiopulmonary examinations. No participants had any diseases and they did not smoke, drink alcohol, or take other medications, which would alter the hormone response. Likewise, to avoid the possible interference of other nutritional supplements on the different variables measured in this investigation, a 2-week washout period was introduced.

All rowers were fully informed of all procedures of the study and signed a statement of informed consent. This research was designed in accordance with the Declaration of Helsinki (2008) and the Fortaleza update (2013) and was approved by the Human Research Ethics Committee at the Basque Country University, Vitoria (M10/2017/247).

### 2.2. Experimental Protocol and Evaluation Plan

This study was designed as a randomized, double-blind, and placebo-controlled study to evaluate the influence of 10-week oral supplementation of CrM plus HMB on EIMD markers and anabolic/catabolic hormones in this sport population.

The participants were randomly assigned to four groups (placebo group (PLG), CrM group (CrMG), HMB group (HMBG), and CrM-HMB group (CrM-HMBG)) by an independent statistician using a stratified block design: PLG (*n* = 7; height: 184.9 ± 2.4 cm and body mass: 81.9 ± 6.3 kg), CrMG (*n* = 7; 0.04 g/kg/day of CrM; height: 183.4 ± 7.8 cm and body mass: 81.2 ± 5.0 kg), HMBG (*n* = 7; 3 g/day of HMB; height: 185.5 ± 10.1 cm and body mass: 79.9 ± 12.2 kg), and CrM-HMBG (*n* = 7; 0.04 g/kg/day of CrM plus 3 g/day of HMB; height: 181.6 ± 4.3 cm and body mass: 78.0 ± 4.7 kg). The CrM and HMB supplementation dosages were selected in accordance with a supplementation standard for athletic performance [34,35] and elite rowers [41]. All participants attended the laboratory at 8:30 a.m. for blood collection at two different times during the intervention period: (1) at baseline (T1) and (2) post-treatment after 10 weeks of supplementation (T2).

All rowers performed six exercise sessions per week during the 10 study weeks (which were exactly the same for all participants), with a duration of 1.5 h/day (distributed as 60% aerobic work on a traditional boat, 30% resistance training in a gym, and 10% complementary training: injury prevention, core stability, and articular mobility). These exercise sessions included two rowing official races or equivalent training per week. To homogenize the competition load, rowers who did not participate in the competitions held a training session with the same load as a competition. The athletes completed a total of 96.6 h of exercise during the study.

All participants took either the placebo or supplements during the 6 days of weekly training mixed with a chocolate recovery shake (1 g/kg of carbohydrates and 0.3 g/kg protein) within half an hour of finishing the exercise [49]. No masking substance was added to the PLG, given that athletes were unaware of the recovery shake content. On the off day, all rowers ingested the same dose of supplements 30 min before going to bed in a chocolate shake provided by an independent nutritionist. The athletes and researchers did not know the supplementation (double-blind) that they were taking because an independent nutritionist from the rowing club made the recovery shakes with the individual supplementation and verified that all rowers complied with the protocol of ingesting the supplements. The CrM was acquired from Creapure^®^ powder and the HMB from HMB-Ca FullGas^®^ (Fullgas Sport, S.L, 20115 Astigarraga, Guipúzcoa, Spain).

### 2.3. Blood Collection

For the evaluation of muscle damage markers and hormonal parameters in T1 and T2, antecubital venous blood samples were collected from all athletes. All samples were taken in basal conditions after, at least, 12 h of fasting and 36 h without exercise. To standardize blood samples in T1 and T2, the rowers performed the same training session in the last training session prior to blood sample collection. This training session consisted predominantly of concentric activity (lower muscle damage) and was of short duration (30 min) during the “regeneration microcycle”. The “regeneration microcycle” is designed to remove fatigue, not only physical, but also mental, and to restore energy that has been used during previous microcycle [50]. On T1 and T2, the rowers arrived at the laboratory at 8:30 a.m. and the blood samples were collected after being at rest for 30 min.

Biochemical EIMD serum markers (AST, CK, and LDH) were measured using Hitachi 917 automatic autoanalyzer (Hitachi Ltd., Tokyo, Japan).

Hormone status was measured in this study (testosterone and cortisol) through different methods. Serum testosterone was measured by commercially available enzyme immunosorbent assay kits (DRG testosterone ELISA kit^®^, DRG Instruments GmbH, Marburg/Lahn, Germany). The intra-assay coefficient of variation (CV) was 4.3% and the CV between the trials was 9.2%. Otherwise, for serum cortisol measurement, an enzyme-linked fluorescent assay with the aid of a multiparametric analyzer (Minividas^®^, Biomerieux, Marcy l’Etoile, France) was used. The substrate 4-methylumbelliferone was used and fluorescence emission was performed at 450 nm, and after stimulation, at 370 nm. The intra-assay CV was 5.7% and the CV of the intermediate assay was 6.2%.

Finally, the T/C was calculated from the results of testosterone and cortisol by dividing testosterone by cortisol.

### 2.4. Anthropometry and Body Composition

The International Society for the Advancement of Kinanthropometry (ISAK) protocol [51] was used for anthropometrical measures and all participants were measured by the same internationally certified anthropometrist (ISAK Level 3) at both T1 and T2.

All measurements were taken twice and if the difference between both measures exceeded 5% for an individual skinfold, a third measurement was taken. For the analysis, when the measurement was taken twice, the mean was chosen, and when three measurements were taken, the median was chosen. Height (cm) was measured using a SECA^®^ measuring rod (Mod. 220; SECA Medical, Bradford, MA, USA), with 1 mm precision, while body mass (BM) (kg) was assessed by a SECA^®^ model scale (Mod. 220; SECA Medical, Bradford, MA, USA), with 0.1 kg precision. Body mass index (BMI) was calculated using the equation BM/height^2^ (kg/m^2^). Girths (cm) (flexed arm, mid-thigh, and calf girth) were measured with a metallic Lufkin^®^ W606PM measuring tape (Cooper Tools, Apex, NC, USA), with a precision of 1 mm. Finally, the sum of six skinfolds (mm) (triceps, subscapular, suprailiac, abdominal, front thigh, and medial calf) was determined by a Harpenden^®^ skinfold caliber (Harpenden Skinfold Caliber, British Indicators Ltd., London, UK), with 0.2 mm precision. Fat mass and muscle mass percentages were predicted using the Carter [52] and Lee [53] equations, respectively.

### 2.5. Dietary Assessment

The team nutritionist informed all rowers about proper food tracking and instructed them on two different validated methods of dietary recall [54]. The first method consisted of completing a food frequency questionnaire (FFQ) at T2, which has been used previously in other athlete populations [55]. The FFQ, which includes 139 different foods and drinks arranged by food type and meal pattern, asked the participants to recall their food intake based on certain “frequency” categories over the previous 10 weeks. Frequency categories consist of how many times an item was consumed per day, week, or month. Daily energy (kcal) and macronutrient (g) consumption was determined by dividing the reported intake by the frequency in days.

The second method was to complete a seven-day dietary recall before T1 and T2 to check if the results of this recall were similar to the FFQ. When rowers weighed the food, the data were used for the recall; however, if it was not possible to weigh the food, serving sizes consumed were estimated through the standard weight of food items or by determining the portion size by looking at a book containing 500 photographs of food.

Food intake was converted to total energy, macronutrient, and micronutrient values by a validated software package (Easy diet^©^, online version). Easy diet^©^ is based on Spanish tables of food composition [56] and was developed by the Spanish Center for Higher Studies in Nutrition and Dietetics (CESNID). Thus, the total energy and macronutrient intake of each participant was calculated per kilogram of individual body mass.

### 2.6. Statistical Analysis

The data are presented as means and standard deviations. Statistical significance was indicated when *p* < 0.05. Differences from T1 to T2 in each group were assessed by paired *t* tests after the normality of the data was established with the Shapiro–Wilk test (*n* < 50) based on parametric or nonparametric data. A two-way repeated measures analysis of variance (ANOVA) test was used to examine interaction effects (time × supplementation group) among supplementation groups (PLG, CrMG, HMBG, and CrM-HMBG) for power output. A Bonferroni post-hoc test was applied for pairwise comparisons among groups. The percentage changes of the variables studied in each study group between the baseline (T1) and post-treatment (T2) tests were calculated as Δ (%): ((T2 − T1)/T1) × 100).

Effect sizes among participants were calculated using partial eta squared (η^2^p). Since this measure is likely to overestimate the effect sizes, the values were interpreted according to Ferguson [57], which is indicated as having no effect if 0 ≤ η^2^p < 0.05, a minimum effect if 0.05 ≤ η^2^p < 0.26, a moderate effect if 0.26 ≤ η^2^p < 0.64, and a strong effect if η^2^p ≥ 0.64 [57].

The following calculation was used to express the variables in conditions CrMG, HMBG, and CrM-HMBG as a percentage change from condition PLG [43]:
Normalized change (%) = (treatment (CrMG, HMBG, or CrM-HMBG)/control (PLG) − 1) × 100.

Using the additive model, stressor (in fact, all variables) interactions were categorized as either synergistic or antagonistic. Significant interactions suggest that the effect size of one variable has been reduced (antagonistic) or accentuated (synergistic) by the presence (or effect) of the other, whereas additive effects are shown during net stressor independence (i.e., no interaction) [43]. Interactions are best illustrated using variables A and B: (1) additive = A and B combined = A + B individually; (2) synergistic = A and B combined > A + B individually; (3) antagonistic = A and B combined < A + B individually; (4) nullifying = A and B combined = A or B individually; (5) multiplicative = A and B combined = A × B individually.

The analyses were performed using SPSS^®^ software version 24.0 (SPSS, Inc., Chicago, Illinois, USA) and Microsoft Excel^®^ version 19 (Microsoft Corporation, Redmond, WA, USA).

## 3. Results

During the study, the athletes did not show significant statistical differences (*p* > 0.05) in energy and macronutrient intake values among groups (Table 1).

Body mass, BMI, muscle mass percentage, and fat mass percentage did not show significant differences (*p* > 0.05) in the group-by-time interaction (Table 2). However, significant decrements (*p* < 0.05) were found between the two study moments for body mass in PLG, CrMG, HMBG, and CrM-HMBG and for BMI in HMBG and CrM-HMBG (Table 2).

Table 3 shows significant differences in the group-by-time interaction for testosterone (*p* = 0.006; η^2^p = 0.454) and T/C (*p* = 0.32; η^2^p = 0.349). Moreover, significant increases between T1 and T2 (*p* < 0.05) were observed for cortisol in PLG, CrM, HMBG, and CrM-HMBG, for testosterone in CrM-HMBG, and for T/C in CrM and HMBG (Table 3).

Figure 1 shows significant differences in testosterone percentage change (*p* = 0.006; η^2^p = 0.457) among PLG and CrMG, HMBG, and CrM-HMBG and among CrM-HMBG and CrMG and HMBG (*p* > 0.05). In addition, T/C percentage change showed statistical differences (*p* = 0.029; η^2^p = 0.399) among CrM-HMBG and PLG as well as CrMG and HMBG. However, there were no significant differences among groups in cortisol percentage change (*p* = 0.568; η^2^p = 0.094).

EIMD markers (AST, LDH, and CK) did not present statistically significant differences in the group-by-time interaction between T1 and T2 (Table 4). However, significant increases between T1 and T2 were observed for AST in PLG (T1: 17.83 ± 2.79 vs. T2: 22.00 ± 1.55).

Figure 2 shows that there were no significant differences in the percentage change of any EIMD markers between T1 and T2 (*p* > 0.05).

Table 5 shows that an antagonistic effect of combined supplementation was found in comparison with the sum of both supplements isolated on cortisol (CrM-HMBG = 131.55% vs. CrMG + HMBG = 389.99%). Likewise, a synergistic effect of combined supplementation was found with respect to the sum of the two supplements separately on testosterone (CrM-HMBG = −63.85% vs. CrMG + HMBG = −37.89%) and on T/C (CrM-HMBG = 680% vs. CrMG + HMBG = 57.68%).

## 4. Discussion

This study was carried out to determine the effect and degree of potentiation of long-term (10 weeks) combination of 3 g/day of HMB plus 0.04 g/kg/day of CrM supplementation on EIMD markers and testosterone and cortisol status in elite male traditional rowers. The main result revealed an increase in testosterone and a better post-treatment T/C ratio in CrM-HMBG compared with PLG, CrMG, and HMBG after 10 weeks of supplementation. Moreover, the combination of CrM plus HMB presented a synergistic effect on testosterone and T/C and an antagonistic effect on cortisol compared with the sum of individual or isolated supplementation (CrM + HMB). However, the results did not display any differences in EIMD markers (AST, CK, and LDH) among the different supplemented groups.

Maximum performance requires an adequate balance between training loads and recovery that allows improving muscle recovery and avoiding fatigue [58]. To control this balance in order to recover and/or avoid fatigue, there are several variables utilized, such as markers of EIMD and anabolic/catabolic hormones [1,6,17,18]. Although there is an acute increase in EIMD markers after exercise, especially with an eccentric component [6], the maintenance of high EIMD values long term could be indicative of an imbalance between load and recovery, which could reflect chronic fatigue or overtraining [1]. In particular, to better understand this balance, several authors have proposed that anabolic/catabolic hormone status is changed after exercise due to an acute effect [13]. However, in the long term, an increase in testosterone level would indicate better endogenous recovery [59], and an increase in cortisol would indicate increased stress and/or fatigue [10]. Therefore, some authors have indicated that the T/C ratio is an objective indicator of the fatigue status of an athlete [15]. Thus, a decrease in this ratio in the long term would indicate greater stress, while an increase would indicate better recovery [14].

In order to recover faster and delay or avoid fatigue, some supplements have been proposed, such as CrM or HMB [19,20]. Given that CrM increases muscle creatine stores, the creatine-phosphocreatine Cr-PCr shuttle is positively affected. One of the main roles of the Cr-PCr system is the cellular energy transport system (the Cr-PCr shuttle). This shuttle presents a higher energy availability during exercise through free Cr speeding up PCr resynthesis in the mitochondria. In the mitochondria outer membrane, Cr reacts with the mitochondrial isoenzyme of creatine kinase (mi-CK) using adenosine triphosphate (ATP) and turning it in adenosine diphosphate (ADP). The ADP is resynthesized into ATP inside the mitochondrial inner membrane/matrix space after reacting with ATP synthase (consuming protons (H^+^)). Therefore, the ATP is reused again in the mi-CK and Cr reaction, restarting the shuttle [60]. On the other hand, HMB can augment mitochondrial biogenesis by an increase in the gene expression of peroxisome proliferator-activated receptor (PGC-1α), which contributes to better oxidative function and hence enhances the aerobic capacity [61]. Mitochondrial biogenesis consists of increasing the quantity and density of muscle cell mitochondria, angiogenesis, and fat oxidation [62]. Therefore, these adaptations could help save glycogen when same training intensities are performed, making them one of the most important limiting factors during exercise [63].

Despite the promising results, the effect on EIMD and hormonal status, and the different metabolic pathways of action of CrM and HMB supplementation, the combination of these supplements has presented controversial results for EIMD and hormone status [64]. These studies did not find differences in CK [44,45,46,47] and LDH [44,47] levels among supplemented groups. In fact, in the study by Jowko et al. (20 g/day for the first seven days and 10 g/day for the rest of the days of CrM and 3 g/day of HMB) [46], CK levels remained elevated following three weeks of CrM/HMBG supplementation, suggesting that CrM antagonized the CK-lowering effect of HMB. However, the results obtained in this study on EIMD did not show an antagonistic effect on EIMD markers. Moreover, there were no differences between T1 and T2 in any of the study groups that could indicate an adequate balance between training loads and recovery [4,20]. These differences could be influenced by blood sampling timing when the objective is to determine the long-term effect of CrM plus HMB supplementation and insufficient time is provided to avoid EIMD acute increases post-exercise [45]. Likewise, these differences might be due to the fact that this type of sport entails short (approximately 20 min) cyclic activity, in contrast to endurance and team sports (high eccentric activity with heavy weights) [44,45,46,47,65] that produce a higher rate of muscle damage [31,66].

Regarding anabolic/catabolic hormones, to our knowledge, only one study has analyzed the anabolic/catabolic hormone status when mixed supplements were ingested. Crowe and O’Connor did not show significant differences in cortisol or testosterone levels after six weeks of CrM plus HMB supplementation despite receiving identical supplementation dosages (3 g/day of CrM and 3 g/day of HMB) [45]. In the present study, although a significant increase in cortisol levels was observed in all study groups between T1 and T2, only a significant increase in testosterone levels was observed in CrM-HMBG. These changes resulted in a significant decrease of the T/C ratio in PLG, CrMG, and HMBG, which indicates greater accumulated stress/fatigue [67]. The differences in testosterone levels could be due to the duration of the intervention (six weeks in Crowe and O’Connor’s study [45] vs. 10 weeks in the present study), given that testosterone requires long-term intervention to produce significant changes [11,12]. Moreover, the results presented on testosterone and T/C could mean better training adaptation/recovery mediated by the combination of CrM and HMB. This result might have facilitated a better aerobic power in CrM-HMBG during an incremental test (related to individual lactate threshold) [68]. These results were corroborated by a synergistic effect of combined supplementation (CrM plus HMB) on testosterone and T/C and an antagonistic effect on cortisol. However, after reviewing the literature, to the best of our knowledge, the mechanisms by which the combination of CrM plus HMB increase testosterone and the T/C ratio are unclear; hence, further investigation is needed.

### 4.1. Limitations, Strengths, and Future Research

The results of this experimental study should be treated with caution due to the small sample size in total (*n* = 28) and in each study group (*n* = 7), which is common in elite sports because it is very difficult to obtain larger samples in this population. However, the methodology of a study is its most important strength. Given that this study was a double-blind, placebo-controlled trial with nutrition control, it was able to avoid possible factors that could influence the CrM plus HMB combination effect. Moreover, as another strength of this study, the diet ingested by the athletes was controlled as well as the body composition throughout the intervention process, so that these parameters did not influence the final results.

Future research should analyze the cellular signal/gene mechanism by which both combined supplements are synergistic on testosterone level. Moreover, it should analyze how this combination affects the female population or anaerobic sports, given that this study only focused on males and measured aerobic performance.

### 4.2. Practical Application

This study could be interesting for nutritionists and physicians who want to provide better post-training or post-competition recovery for their athletes. Taking into account that 0.04 g/kg/day of CrM plus 3 g/day of HMB for 10 weeks could improve muscle and endogenous recovery, supplementation phases could be considered in the training phases in which there is a greater load.

## 5. Conclusions

In summary, the combination of 3 g/day of CrM plus 0.04 g/kg/day of HMB for 10 weeks showed an increase in testosterone and T/C compared with placebo or isolated supplementation. Moreover, this combined supplementation revealed a synergistic effect on testosterone and T/C and an antagonistic effect on cortisol, which are positive results for athletes’ recovery. However, this combination did not present any differences in EIMD. Therefore, the combined use of these two ergogenic supplements could promote faster muscle recovery from high-intensity activity but without preventing muscle damage.

## Figures and Tables

**Figure 1 biomolecules-10-00140-f001:**
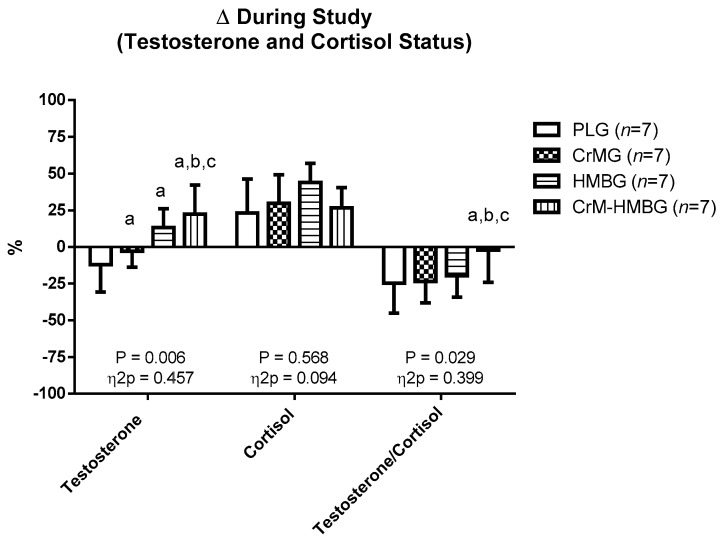
Percentage changes during the study in cortisol and testosterone hormone status and testosterone/cortisol ratio in the four study groups. Data are expressed as mean ± standard error. Δ: ((T2 − T1)/T1) × 100; differences among groups in each test by ANOVA test (*p* < 0.05): a: regarding PLG; b: regarding CrMG; c: regarding HMBG.

**Figure 2 biomolecules-10-00140-f002:**
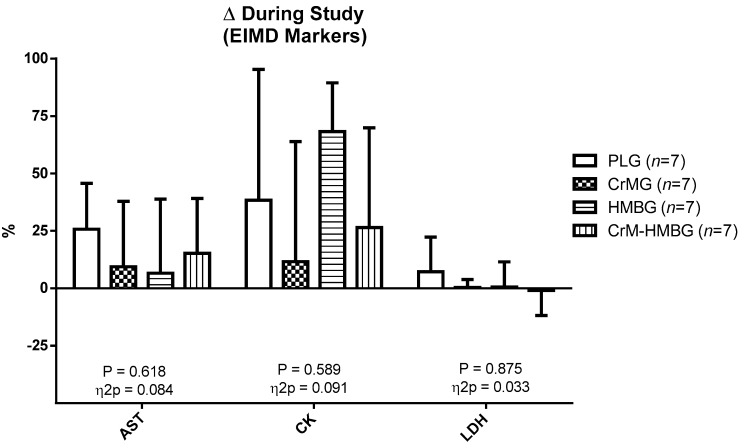
Percentage changes in exercise-induced muscle damage (EIMD). Data are expressed as mean ± standard error. Δ: ((T2 − T1)/T1) × 100; differences among groups in each test by ANOVA test.

**Table 1 biomolecules-10-00140-t001:** Energy and macronutrient intake in each study group during 10 weeks of study.

	PLG	CrMG	HMBG	CrM-HMBG
Energy (kcal/kg)	44.8 ± 6.2	45.0 ± 6.6	44.7 ± 6.3	45.1 ± 7.0
Protein (g/kg)	1.9 ± 0.4	2.0 ± 0.6	1.9 ± 0.7	1.9 ± 0.4
Fat (g/kg)	1.5 ± 0.4	1.6 ± 0.5	1.5 ± 0.6	1.6 ± 0.6
Carbohydrates (g/kg)	6.0 ± 0.9	6.1 ± 1.1	6.1 ± 1.3	6.0 ± 1.2

Data are expressed as mean ± standard deviation. PLG: placebo group, CrMG: creatine monohydrate group, HMBG: β-hydroxy β-methylbutyrate group.

**Table 2 biomolecules-10-00140-t002:** Anthropometry and body composition of participants.

Group	T1	T2	P (T × G)	η^2^p
Body mass (kg)				
PLG	81.9 ± 6.3	80.0 ± 5.3 *	0.883	0.028
CrMG	81.2 ± 5.0	78.6 ± 5.4 *		
HMBG	79.9 ± 12.2	77.6 ± 11.1 *		
CrM-HMBG	78.0 ± 4.7	75.5 ± 4.5 *		
BMI (kg/m^2^)				
PLG	24.1 ± 2.4	23.6 ± 2.0	0.951	0.016
CrMG	24.1 ± 1.5	23.4 ± 1.6 *		
HMBG	24.1 ± 1.8	23.3 ± 1.8 *		
CrM-HMBG	23.7 ± 1.6	22.9 ± 1.4 *		
Muscle mass (%)				
PLG	40.3 ± 2.5	41.0 ± 2.3	0.789	0.104
CrMG	40.6 ± 2.4	41.7 ± 2.7		
HMBG	41.1 ± 2.0	41.7 ± 2.2		
CrM-HMBG	41.2 ± 2.4	42.1 ± 2.3		
Fat mass (%)				
PLG	8.9 ± 1.5	8.7 ± 1.4	0.884	0.030
CrMG	8.6 ± 1.7	8.4 ± 1.8		
HMBG	8.7 ± 1.5	8.4 ± 1.1		
CrM-HMBG	8.5 ± 1.6	8.2 ± 1.4		

Data are expressed as mean ± standard deviation. P (T × G): group-by-time interaction (*p* < 0.05, all such occurrences). Two-factor repeated-measures ANOVA. * Significantly different between phases (T1 vs. T2); *p* < 0.05. BMI: body mass index.

**Table 3 biomolecules-10-00140-t003:** Participant’s testosterone and cortisol hormone status.

Group	T1	T2	P (T × G)	η^2^p
Testosterone (ng/dL)				
PLG	5.22 ± 0.56	4.56 ± 0.84	0.006	0.454
CrMG	4.27 ± 0.73	4.20 ± 1.12		
HMBG	4.90 ± 0.95	5.60 ± 1.56		
CrM-HMBG	4.91 ± 0.87	5.97 ± 1.23 *		
Cortisol (μg/dL)				
PLG	15.87 ± 2.99	17.93 ± 2.08 *	0.451	0.121
CrMG	15.75 ± 2.78	20.80 ± 6.13 *		
HMBG	16.32 ± 1.29	23.30 ± 2.97 *		
CrM-HMBG	18.18 ± 1.13	22.95 ± 1.89 *		
Testosterone/cortisol ratio				
PLG	33.77 ± 12.03	25.97 ± 6.58 *	0.032	0.349
CrMG	28.04 ± 7.49	22.17 ± 9.90 *		
HMBG	30.19 ± 6.33	23.94 ± 4.98 *		
CrM-HMBG	27.07 ± 5.12	26.07 ± 5.29		

Data are expressed as mean ± standard deviation. P (T × G): group-by-time interaction (*p* < 0.05, all such occurrences). Two-way repeated-measures ANOVA. * Significantly different between two phases (T1 vs. T2), *p* < 0.05.

**Table 4 biomolecules-10-00140-t004:** Exercise-induced muscle damage markers in the four study groups at baseline (T1) and after 10 weeks (T2).

Group	T1	T2	P (T × G)	η^2^p
AST (UI/L)				
PLG	17.83 ± 2.79	22.00 ± 1.55*	0.648	0.077
CrMG	22.33 ± 9.65	23.50 ± 8.89		
HMBG	18.00 ± 3.29	19.00 ± 6.96		
CrM-HMBG	21.33 ± 4.68	24.67 ± 7.30		
CK (UI/L)				
PLG	190.50 ± 94.79	216.83 ± 97.35	0.641	0.079
CrMG	277.00 ± 171.40	256.33 ± 130.38		
HMBG	147.50 ± 63.91	243.33 ± 286.42		
CrM-HMBG	201.67 ± 105.64	260.50 ± 159.28		
LDH (UI/L)				
PLG	293.17 ± 36.23	310.17 ± 22.98	0.792	0.049
CrMG	337.00 ± 47.69	337.33 ± 39.20		
HMBG	340.17 ± 27.64	342.17 ± 45.35		
CrM-HMBG	339.17 ± 24.77	337.83 ± 60.72		

Data are expressed as mean ± standard deviation. P (T × G): group-by-time interaction (*p* < 0.05, all such occurrences). Two-way repeated-measures ANOVA. * Significantly different between two phases (T1 vs. T2), *p* < 0.05. AST: aspartate aminotransferase, CK: creatine kinase, LDH: lactate dehydrogenase.

**Table 5 biomolecules-10-00140-t005:** Determination of the effect of the combination of supplements.

Group	% Change to Placebo	Effect
Cortisol		
CrMG	145.15%	Antagonistic 131.55 < 389.99
HMBG	238.84%	
CrMG + HMBG	389.99%	
CrM-HMBG	131.55%	
Testosterone		
CrMG	4.21%	Synergistic −63.85 > −37.85
HMBG	−42.1%	
CrMG + HMBG	−37.89%	
CrM-HMBG	−63.85%	
Testosterone/cortisol ratio		
CrMG	32.88%	Synergistic 680 > 57.68
HMBG	24.80%	
CrMG + HMBG	57.68%	
CrM-HMBG	680%	

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
