# Peer review of "Long-Term Effect of Combination of Creatine Monohydrate Plus β-Hydroxy β-Methylbutyrate (HMB) on Exercise-Induced Muscle Damage and Anabolic/Catabolic Hormones in Elite Male Endurance Athletes"

_biomolecules, 2020, doi:10.3390/biom10010140_

Round 1

Reviewer 1 Report

Table 1 : anthropometry. For the CrM-HMBG gruop a decrease in body mass 78.0±4.7 75.5±4.5* was found. However, this differences is not supported in the same table by the changes in muscle mass and fat mass. Thus the methods used seems not very accurate or some errors are in the table. The caption of the table is redunant in reporting again the table contents. 

In table 3 is reported a decrease in T/C ratio : CrM-HMBG 27.07±5.12 vs 26.07±5.29, while in the discussion (rows 287and 288) is stated:

"The main result described an increase on testosterone and T/C ratio in CrM-HMBG compared to PLG, CrMG and HMBG after 10 weeks of supplementation."

This statement is in disagreement with the data reported in the table.

In the caption of table 3, data reported in the table are un-usefulness report again.

Author Response

Dear Reviewer,

We really appreciate the time you devoted to reading our manuscript and helping us to craft an improved version. We are pleased to clarify your concern which, we believe will improve the impact and quality of your work. Please find below our response to your observation. We have made a concerted attempt to systematically address the specific concerns raised for this revision and we have highlighted the alterations to this revision within the manuscript in yellow for your convenience.

In Advanced,

King Regards

Reviewer 1

Reviewer #1: Comments and Suggestions for Authors

REVIWER: Table 1: anthropometry. For the CrM-HMBG group a decrease in body mass 78.0±4.7 75.5±4.5* was found. However, this difference is not supported in the same table by the changes in muscle mass and fat mass. Thus, the methods used seem not very accurate or some errors are in the table. The caption of the table is redundant in reporting again the table contents.

AUTHORS: Thank you for this comment. You are right. For that reason, we inserted an erroneous transcription information inside the anthropometry and body composition table. We have corrected this information in the table.

On the other hand, to avoid redundant information based on your suggestion, we have changed the caption of the table 1.

REVIWER: In table 3 is reported a decrease in T/C ratio: CrM-HMBG 27.07±5.12 vs 26.07±5.29, while in the discussion (rows 287and 288) is stated:

"The main result described an increase on testosterone and T/C ratio in CrM-HMBG compared to PLG, CrMG and HMBG after 10 weeks of supplementation."

This statement is in disagreement with the data reported in the table.

AUTHORS: Agree, thank you for the comment. We have changed it in the manuscript.

"The main result revealed an increase in testosterone and a better post-treatment T/C ratio in CrM-HMBG compared with PLG, CrMG, and HMBG after 10 weeks of supplementation"

REVIWER: In the caption of table 3, data reported in the table are un-usefulness report again.

AUTHORS: Thank you for the comment. We agree with you and we have removed the un-usefulness data of the table 3 caption.

Reviewer 2 Report

The present manuscript examines the effect of supplementations with creatine and HMB, combined or alone, during a 10-week training period on muscle damage markers and hormonal levels. In spite of interesting, the manuscript presents some important flaws.

General comment

English should be carefully and extensively revised.

Specific main comments

The rationale for expecting a synergistic effect of the supplemented creatine and HMB is not clearly indicated.

Rationale for the chosen dosage (amount of both supplements) should be included. Effects of the supplementation in terms of, for example, increasing creatine muscle levels, increasing HMB levels, or any other similar could be discussed. Why one of the lowest creatine supplementation protocols, in terms of quantity, was used?

Did athletes participate in competitions during this 10-week period? Was the study performed during the pre-season? Please, clarify.

More details about the training sessions should have been provided. Intensity, type of exercises performed, etc. Any measurement of physical activity performed by participants during training sessions? Did all the participants complete all training sessions? Any drop-outs during the study? It could be expected that in a 10-week period of training some of the participants become injured.

Any measurement of the participants’ fitness? For example, was any VO2max test, or similar performed?

Any measurement of total physical activity performed by participants during the study?

In spite of it is indicated as a limitation, I should say that 7 per group is a very small sample size. Was a previous sample size calculation performed?

Does all participants belong to the same team? Actually authors indicated, in line 352, that there are only 18 rowers in each team. Therefore, can the authors ensure that all participants were under the same conditions? Were these training sessions the same for all participants? Were the training sessions’ intensity adjusted for each subject?

Authors indicate that blood samples were taken after 36 hours without exercise (line 140). However, it is also indicated that muscle damage markers, among others, could be increased until 24-96 hours (line 57), and I should say that even longer. Therefore, measuring the T2 markers after only 36 hours without performing exercise cannot be considered adequate at all because this supposes a too short time as it is indicated by the authors. This should be clearly clarified.

Did the authors consider measuring the acute effects during a single training session? In spite of this was not the aim of the study, this measurement could have allowed to clarify questions such as the last one indicated as well as enhanced relevance of the manuscript.

Did the authors consider taking an intermediate (after 5 weeks) blood sample? This would have been useful increasing understanding of results.

Results section. Prevent the repetition of results showed in tables and figures in the text. Remove all values from the text, highlighting only the most important result from each table or figure and the statistical significances. Otherwise tables and figures could be removed.

In my opinion T1 cortisol values should be carefully taken and/or revised. The T1 values from the CrM-HMBG are about 30% higher than in the PLG. Any reason for this? This suppose a very important different between groups at the beginning of the study, actually higher than the difference between the T1 and T2 values in this group. This is particularly relevant because the main result highlighted at the beginning of the discussion (increased ratio) is influenced by this cortisol value.

Discussion. There’s too much information repeated in the introduction and the discussion. Authors should reduce and limit these repetitions.

Additional specific comments

Rationale for considering total proteins as EIMD marker should be included as it is not a common usage of this parameter (or references supporting this validity).

Line 177. Explain “before the start of T1 and T2”. These T1 and T2 are supposed to be simple time points, aren’t they? Thus, what is the meaning of “start”?

How was the supplementation in the resting day taken? Was the same shake used? How was this resting-day supplement provided to all participants? Who did provide this supplement?

Consider removing figure 1 and 2, as these suppose repeated information and given that the most important information is the one from the absolute values.

Line 301. The sentence included seems to be incomplete.

Discussion. Please, provide dosage used in previous discussed studies to allow a proper interpretation of results.

Conclusions. Consider adding at the end “but without preventing muscle damage”.

Author Response

Dear Reviewer,

We really appreciate the time you devoted to reading our manuscript and helping us to craft an improved version. We are pleased to clarify your concern which we believe will improve the impact and quality of your work. Please find below our response to your observation. We have made a concerted attempt to systematically address the specific concerns raised for this revision and we have highlighted the alterations to this revision within the manuscript in yellow for your convenience.

In advanced,

King Regards

Reviewer(s)' Comments to Author:

Reviewer 2

Reviewer #2: Comments and Suggestions for Authors

General comment:

REVIEWER: English should be carefully and extensively revised.

AUTHOR: Thanks so much for your help. The manuscript has undergone English language editing by MDPI. The text has been checked for correct use of grammar and common technical terms, and edited to a level suitable for reporting research in a scholarly journal.

Specific main comments:

REVIEWER: The rationale for expecting a synergistic effect of the supplemented creatine and HMB is not clearly indicated.

AUTHORS: Thank you for your observation. We have added in the introduction section, concrete information about Creatine and HMB pathways in order to expect better synergistic effect of the supplemented creatine plus HMB.

REVIEWER: Rationale for the chosen dosage (amount of both supplements) should be included. Effects of the supplementation in terms of, for example, increasing creatine muscle levels, increasing HMB levels, or any other similar could be discussed. Why one of the lowest creatine supplementation protocols, in terms of quantity, was used?

AUTHORS: Thank you for your question. We have added the explanation in the experimental protocol and evaluation plan section, explained the reason to the chosen amount of each supplement, based on current positions stands about this topic: “The CrM and HMB supplementation dosages were selected in accordance with a supplementation standard for athletic performance [34, 35] and elite rowers [41].”

On the other hand, the amount of CrM used was included based on specific scientific review for rowers’ supplementation by Boegman and Dziedzic, wich explains this question about CrM supplementation: “Rapid (20 g/day for 5 day) and slow (3 g/day for 28 day) loading protocols with CrM appear equally effective in achieving supramaximal muscle creatine stores, allowing for protocol individualization based on athlete preference” [1].

[1] Boegman, S.; Dziedzic, C. Nutrition and supplements for elite open-weight rowing. Curr. Sports Med. Rep. 2016, 15, 252–261.

Moreover, we decided to avoid the load supplementation option in order to describe an easier double-blind process.

REVIEWER: Did athletes participate in competitions during this 10-week period? Was the study performed during the pre-season? Please, clarify.

AUTHORS: Thank you for this consideration. Yes, the participants took part in competitions during the study period, given that they were in-season period (competitive period). We have added these sentences in order to clarify this question in participants and Experimental protocol and evaluation plan sections, respectively: “These rowers performed the same loads and number of training sessions with a daily duration of 1.5 h/day, 6 days/week for 10 weeks during the in-season period (competitive period).”, and “These exercise sessions included two rowing official races or equivalent training per week.”

REVIEWER: More details about the training sessions should have been provided. Intensity, type of exercises performed, etc. Any measurement of physical activity performed by participants during training sessions? Did all the participants complete all training sessions? Any drop-outs during the study? It could be expected that in a 10-week period of training some of the participants become injured.

AUTHORS: Thank you for the comment. More details have been added about training session in experimental protocol and evaluation plan section in material and methods section:

In particular, “All rowers performed six exercise sessions per week during the 10 study weeks (which were exactly the same for all participants), with a duration of 1.5 h/day (distributed as 60% aerobic work on a traditional boat, 30% resistance training in a gym, and 10% complementary training: injury prevention, core stability, and articular mobility). These exercise sessions included two rowing official races or equivalent training per week. The athletes completed a total of 96.6 h of exercise during the study.”

On the other hand, all of 28 rowers completed  the study period without any significant injury (included in the text). In this way, we have added next sentence inside the participants section: “Although some rowers had minor injuries and could miss some training sessions, all of them completed the entire study rigorously without any drop-out.” 

REVIEWER: Any measurement of the participants’ fitness? For example, was any VO2max test, or similar performed?

AUTHORS: Thanks for your interest. The VO2 max values obtained by incremental test have been added inside the material and methods section (description of participants) (VO2 max:59.92 mL/min/kg).

On the other hand, the rowers performed an incremental test in T1 and T2 to determinate the power and the training zones. These results are under review in another MDPI journal (nutrients-680408- “Effect of 10 weeks of creatine monohydrate plus HMB supplementation on athletic performance test in elite male endurance athletes”).

REVIEWER: Any measurement of total physical activity performed by participants during the study?

AUTHORS: Thank you for the comment. Please see previous answer where we have explained training sessions.

REVIEWER: In spite of it is indicated as a limitation, I should say that 7 per group is a very small sample size. Was a previous sample size calculation performed?

AUTHORS: Thank you very much for your interest. The total sample was obtained by the total rowers of 1 of the 12 teams that make up the rowing First league. To clarify this issue, next sentence is added: “Twenty-eight elite male traditional rowers (30.43 ± 4.65 years and 59.92 mL/min/kg of VO2 max) belonging to 1 of the 12 teams that make up the rowing First Trainer League in Spain (ACT) participated in this study.”

We divided the team into 4 groups in order to carry out an adequate protocol. However, we have added this sentence in Limitations, Strengths and Future Research section: “The results of this research should be treated with caution due to the small sample size in total (n = 28) and in each study group (n = 7), which is common in elite sports because it is very difficult to obtain larger samples in this population.”

we divide the team into 4 groups to ensure a proper protocol

REVIEWER: Does all participants belong to the same team? Actually, authors indicated, in line 352, that there are only 18 rowers in each team. Therefore, can the authors ensure that all participants were under the same conditions? Were these training sessions the same for all participants? Were the training sessions’ intensity adjusted for each subject?

AUTHORS: Thank you for the comment. It has been a writing mistake, in the teams are 28 rowers and we have corrected it thought the text.

On the other hand, as we have explained in a previous comment, the rowers performed 2 tests to determine the individual training zones. However, the practice load sessions (intensity and duration) were exactly the same for all participants. In this sense, we have included this information in participants section: “These rowers performed the same loads and number of training sessions with a daily duration of 1.5 h/day, 6 days/week for 10 weeks during the in-season period (competitive period).

REVIEWER: Authors indicate that blood samples were taken after 36 hours without exercise (line 140). However, it is also indicated that muscle damage markers, among others, could be increased until 24-96 hours (line 57), and I should say that even longer. Therefore, measuring the T2 markers after only 36 hours without performing exercise cannot be considered adequate at all because this supposes a too short time as it is indicated by the authors. This should be clearly clarified.

AUTHORS: Thank you for your comment. The increases in muscle damage markers are shown until 24-96 hours or even longer, as you commented. But this increase is dependent on the type of exercise and duration. In this case, for to standardize blood samples in T1 and T2 the rowers performed the same training session in the last day previous of blood samples collection. This exercise session consisted in a predominantly concentric (lower muscle damage) activity and it is neither a long duration event (~30 minutes). To clarify this question the authors have added the next sentence in blood collection section: “To standardize blood samples in T1 and T2, the rowers performed the same training session in the last training session prior to blood sample collection. This training session consisted predominantly of concentric activity (lower muscle damage) and was of short duration (30 min).”

REVIEWER: Did the authors consider measuring the acute effects during a single training session? In spite of this was not the aim of the study, this measurement could have allowed to clarify questions such as the last one indicated as well as enhanced relevance of the manuscript.

AUTHORS: Thank you for the comment. As the reviewer indicated the main objective of our study, it was to determine the effect of these 2 supplements in the long term. Different investigations have shown already that this combination from 6 days to 6 weeks have presented controversial results as indicated throughout of a recent systematic review [1].

[1] Fernández-Landa, J.; Calleja-González, J.; León-Guereño, P.; Caballero-García, A.; Córdova, A.; Mielgo-Ayuso, J. Effect of the combination of creatine monohydrate plus HMB supplementation on sports performance, body composition, markers of muscle damage and hormone status: a systematic review. Nutrients 2019, 11, E2528.

REVIEWER: Did the authors consider taking an intermediate (after 5 weeks) blood sample? This would have been useful increasing understanding of results.

AUTHORS: Thank you for the comment. In the same line that previous answer, the main objective of our study was to determine the effect of these 2 supplements in the long term because different investigations have already used this combination in studies lasting up to 6 weeks with controversial results as indicated throughout the text and published in recent systematic review [1].

[1] Fernández-Landa, J.; Calleja-González, J.; León-Guereño, P.; Caballero-García, A.; Córdova, A.; Mielgo-Ayuso, J. Effect of the combination of creatine monohydrate plus HMB supplementation on sports performance, body composition, markers of muscle damage and hormone status: a systematic review. Nutrients 2019, 11, E2528.

REVIEWER: Results section. Prevent the repetition of results showed in tables and figures in the text. Remove all values from the text, highlighting only the most important result from each table or figure and the statistical significances. Otherwise tables and figures could be removed.

AUTHORS: Thank you very much for your comment. We have removed the un-usefulness information from the tables and figures caption or from the text.

REVIEWER: In my opinion T1 cortisol values should be carefully taken and/or revised. The T1 values from the CrM-HMBG are about 30% higher than in the PLG. Any reason for this? This suppose a very important different among groups at the beginning of the study, actually higher than the difference between the T1 and T2 values in this group. This is particularly relevant because the main result highlighted at the beginning of the discussion (increased ratio) is influenced by this cortisol value.

AUTHORS: Thank you for the comment. Cortisol baseline values of PLG have been corrected. There was a transcription error. Sorry for that mistake, the appropriate value is 15.87±2.99 μg/dL. In this sense, there were no differences between cortisol values at the beginning of the study among the study groups.

REVIEWER: Discussion. There’s too much information repeated in the introduction and the discussion. Authors should reduce and limit these repetitions.

AUTHORS: Thank you for your observation. The authors have deleted information repeated in introduction or discussion and revised both parts.

Additional specific comments:

REVIEWER: Rationale for considering total proteins as EIMD marker should be included as it is not a common usage of this parameter (or references supporting this validity).

AUTHORS: Thank you for the comment. We have removed all data about total protein in the study after your suggestions.

REVIEWER: Line 177. Explain “before the start of T1 and T2”. These T1 and T2 are supposed to be simple time points, aren’t they? Thus, what is the meaning of “start”?

AUTHORS: Thank you for your comment. To avoid misunderstandings, we have erased “the start” in that sentence. The new sentence is: “The second method was to complete a seven-day dietary recall before T1 and T2 to check if the results of this recall were similar to the FFQ.”

REVIEWER: How was the supplementation in the resting day taken? Was the same shake used? How was this resting-day supplement provided to all participants? Who did provide this supplement?

AUTHORS: Thank you for the comment. We have changed the following sentence to clarify the supplementation protocol inside Experimental protocol and evaluation plan section: “During the off- day, all rowers ingested the same dose of supplements, 30 minutes before going to bed in a chocolate shake provided by an independent nutritionist”.

REVIEWER: Consider removing figure 1 and 2, as these suppose repeated information and given that the most important information is the one from the absolute values.

AUTHORS: Thank you for your proposal. Although the most important information is the one from the absolute values, the percentage change values provide complementary information that can facilitate the reader in a more visual way to understand the results. However, if the reviewer considers that this information is not useful, we have not problem to delete these figures.

REVIEWER: Line 301. The sentence included seems to be incomplete.

AUTHORS: Thank you for your comment. The word “while” have been removed from that sentence.

REVIEWER: Discussion. Please, provide dosage used in previous discussed studies to allow a proper interpretation of results.

AUTHORS: Thank you for your comment. The dosages of the discussed studies are now in the discussion.

REVIEWER: Conclusions. Consider adding at the end “but without preventing muscle damage”.

AUTHORS: Thank you for your comment. We have added “but without preventing muscle damage” at the end of the conclusion.

Round 2

Reviewer 2 Report

The authors have improved a lot the present manuscript. However, I think that a few essential questions should be still considered.

Authors indicate that “the rowers performed an incremental test in T1 and T2 to determinate the power and the training zones. These results are under review in another MDPI journal (nutrients-680408- “Effect of 10 weeks of creatine monohydrate plus HMB supplementation on athletic performance test in elite male endurance athletes”).In my opinion this data should be included in the present manuscript, adding value to results shown and, also, preventing the division of results from the same study.

When authors are asked whether participants belong to the same team, the response is that “It has been a writing mistake, in the teams are 28 rowers and we have corrected it thought the text.” This is quite confusing for me because no more than 21-22 rowers per team are shown in the website of the rowing First Trainer ACT League. Furthermore, as the study was performed during the season, competitions become also important. In this sense, it seems that up to 14 rowers participate in each competition, thus it could be differences in the physical activity performed by the 28 participants that is not explained by the authors. In addition, it is not clear whether the role of all rowers participating in the competition, and thus, exercise performed, is the same in the competition.

In order to avoid effects of the previous days training, authors indicate that in T1 and T2 the rowers performed the same training session in the last day previous of blood samples collection. This exercise session consisted in a predominantly concentric (lower muscle damage) activity and it is neither a long duration event (~30 minutes)”. However, clearly, this is not enough to prevent effects of last days training (up to 96 hours as the authors say again in their response). This sampling and timing schedule remains unexplained and, thus, unacceptable.

Authors indicate that “Cortisol baseline values of PLG have been corrected. There was a transcription error.” This is, at least, confusing because a simple transcription mistake in the table should not have induced a change in the value of the testosterone to cortisol ratio (not highlighted) in the same table (from 39.77 to 33.77). Two transcription mistakes in these values are not acceptable. Furthermore, the percentage of change in figure 1 has not been modified, and supposes another mistake. This issue should be clearly clarified.

Author Response

Point-by-Point Response to Reviewer’s Comments

We would like to sincerely thank the reviewer for his/her helpful recommendations again. We have seriously considered all the comments and carefully revised the manuscript accordingly. Revisions are highlighted in green through the manuscript to indicate where changes have taken place. We feel that the quality of the manuscript has been significantly improved with these modifications and improvements based on the reviewers’ suggestions and comments. We hope our revision will lead to an acceptance of our manuscript for publication Biomolecules.

In advance,

King regards

REVIEWER: The authors have improved a lot the present manuscript. However, I think that a few essential questions should be still considered.

AUTHORS: Thank you very much. We have improved the manuscript due to reviewers thoughtful and constructive comments.

REVIEWER: Authors indicate that “the rowers performed an incremental test in T1 and T2 to determinate the power and the training zones. These results are under review in another MDPI journal (nutrients-680408- “Effect of 10 weeks of creatine monohydrate plus HMB supplementation on athletic performance test in elite male endurance athletes”).” In my opinion this data should be included in the present manuscript, adding value to results shown and, also, preventing the division of results from the same study.

AUTHORS: Thank you for this comment. The article “Effect of 10 weeks of creatine monohydrate plus HMB supplementation on athletic performance test in elite male endurance athletes” has been published in Nutrients on January 10. However, we have added this reference to explain that the changes in anabolic-catabolic hormones could has led in an increase on aerobic power values measured by an incremental test: “This result might have facilitated a better aerobic power in CrM-HMBG during an incremental test (related to individual lactate threshold) [68].”

Fernández-Landa, J.; Fernández-Lázaro, D.; Calleja-González, J.; Caballero-García, A.; Córdova, A.; León-Guereño, P.; Mielgo-Ayuso, J. Effect of ten weeks of creatine monohydrate plus hmb supplementation on athletic performance tests in elite male endurance athletes. Nutrients 2020, 12, 193.

REVIEWER: When authors are asked whether participants belong to the same team, the response is that “It has been a writing mistake, in the teams are 28 rowers and we have corrected it thought the text.” This is quite confusing for me because no more than 21-22 rowers per team are shown in the website of the rowing First Trainer ACT League. Furthermore, as the study was performed during the season, competitions become also important. In this sense, it seems that up to 14 rowers participate in each competition, thus it could be differences in the physical activity performed by the 28 participants that is not explained by the authors. In addition, it is not clear whether the role of all rowers participating in the competition, and thus, exercise performed, is the same in the competition.

AUTHORS: Thank you for your comment. The components of the 1st team are 20-22 rowers as shown in the website of the rowing First Trainer ACT League. However, there are 6 more rowers from the “reserve team” that carried out the same training sessions like the official participants and took part in the study. In this line, we included in participants section next sentence: “Twenty-eight elite male traditional rowers (30.43 ± 4.65 years and 59.92 mL/min/kg of VO2 max) belonging to 1 of the 12 teams that make up the rowing First Trainer League in Spain (ACT) participated in this study (22 rowers of the 1st team and 6 of the reserve team).” Moreover, in order to avoid misunderstanding, the authors have deleted the following sentence in Limitations, Strengths, and Future Research section: “In addition, the First Trainer Rowing League in Spain (ACT) consists of 12 teams of 28 rowers each team.”

On the other hand, with the aim to equalize the competition time of the rowers that took part in study, the rowers that did not participate in the competitions realized a simulate competition training session with the same load as a competition. In fact, we have included the next sentence in participants section: “For homogenize the competition load, rowers who did not participate in the competitions held a training session with the same load as a competition.”

REVIEWER: In order to avoid effects of the previous days training, authors indicate that “in T1 and T2 the rowers performed the same training session in the last day previous of blood samples collection. This exercise session consisted in a predominantly concentric (lower muscle damage) activity and it is neither a long duration event (~30 minutes)”. However, clearly, this is not enough to prevent effects of last days training (up to 96 hours as the authors say again in their response). This sampling and timing schedule remain unexplained and, thus, unacceptable.

AUTHORS: Thank you for this nice detail. It is true that the time elapsed between the last day training and the blood samples collection could be not enough to prevent the effects of training on EIMD values (36 h). However, at both times (T1 and T2) all participants performed the same training on the last week prior to the blood samples collection. Therefore, although the absolute values of these variables could be modified according to the time elapsed between the last training and the last days training collection, the behavior of these variables in the 10 weeks (time * group) should not be altered. Likewise, at both times the samples collected were carried out after the same training and 36 hours of rest during the “Regeneration microcycle”. This microcylce is designed to remove fatigue, not only physical, but also mental, and to restore energy that has been used during previous microcycles. In this sense, we have included the next sentence in blood collection section: “To standardize blood samples in T1 and T2, the rowers performed the same training session in the last training session prior to blood sample collection. This training session consisted predominantly of concentric activity (lower muscle damage) and was of short duration (30 min) during the “regeneration microcycle”. The “regeneration microcycle” is designed to remove fatigue, not only physical, but also mental, and to restore energy that has been used during previous microcycle [50].”

Lastly, athletes are related frequently high values in EIMD markers, given that during sports preparation period, it is practically impossible to get 96 hours of total rest. In our case, the traditional rowers are elite athletes who, (with the exception of the holiday period), have 36-48 hours maximum period to rest.

Bosquet, L., Montpetit, J., Arvisais, D., & Mujika, I. (2007). Effects of tapering on performance: a meta-analysis. Medicine & Science in Sports & Exercise, 39(8), 1358-1365.

REVIEWER: Authors indicate that “Cortisol baseline values of PLG have been corrected. There was a transcription error.” This is, at least, confusing because a simple transcription mistake in the table should not have induced a change in the value of the testosterone to cortisol ratio (not highlighted) in the same table (from 39.77 to 33.77). Two transcription mistakes in these values are not acceptable. Furthermore, the percentage of change in figure 1 has not been modified, and supposes another mistake. This issue should be clearly clarified.

AUTHORS: Thanks so much for this detail. It is embarrassing to have to say that in both cases there was a problem of transcription in one of the databases that we used for the descriptive results. This caused a big error. However, the statistics (table and figure) have not presented mistakes, given that the obtained data were extracted from the original correct data base.

The results of the figure 1 have been corrected.